# The Changes in Visual Acuity Values of Japanese School Children during the COVID-19 Pandemic

**DOI:** 10.3390/children9030342

**Published:** 2022-03-02

**Authors:** Shingo Noi, Akiko Shikano, Natsuko Imai, Fumie Tamura, Ryo Tanaka, Tetsuhiro Kidokoro, Mari Yoshinaga

**Affiliations:** 1Research Institute for Health and Sport Science, Nippon Sport Science University, Tokyo 158-8508, Japan; shikano.a@nittai.ac.jp (A.S.); r-tanaka@ouhs.ac.jp (R.T.); kidokoro@nittai.ac.jp (T.K.); 2Doctoral Program in Health and Sport Science, Nippon Sport Science University, Tokyo 158-8508, Japan; n-enomoto@nittai.ac.jp; 3Masterl Program in Health and Sport Science, Nippon Sport Science University, Tokyo 158-8508, Japan; ft72tamura@gmail.com; 4School of Health and Sport Sciences, Osaka University of Health and Sport Sciences, Osaka 590-0496, Japan; 5Faculty of Pharmaceutical Science, Showa Pharmaceutical University, Tokyo 194-8543, Japan; yosinaga@ac.shoyaku.ac.jp

**Keywords:** new coronavirus infection, myopia, screen time, children, Japan

## Abstract

The coronavirus disease 2019 (COVID-19) pandemic may result in a greater decrease in visual acuity (VA) among Japanese children. Our study aimed to examine Japanese children’s VA during the pandemic. VA data were collected using standard eye tests during school health check-ups conducted in 2019 and 2020 on 5893 children, in seven public elementary schools and four public junior high schools in Tokyo, Saitama, Kanagawa, and Shizuoka. VA changes were statistically analyzed. The relationship between the survey year and poor VA yielded a significant regression coefficient for the surveyed years in elementary and junior high school students. The 2019 VA value and VA change from 2019 to 2020 demonstrated a significant regression coefficient in elementary school students with VAs of “B (0.7–0.9)” and “C (0.3–0.6)”, and junior high school students with VAs of “B”, “C”, and “D (<0.3)”. An analysis of the relationship between the survey year and eye laterality of VA yielded a significant regression coefficient in the surveyed years for elementary (OR, 1.516; 95% CI, 1.265–1.818) and junior high school students (OR, 1.423; 95% CI, 1.136–1.782). Lifestyle changes during the COVID-19 pandemic might have affected VA and eye laterality in Japanese children.

## 1. Introduction

On 11 March 2020, the World Health Organization announced that the new coronavirus disease 2019 (COVID-19) had become a global pandemic [1]. This pandemic has changed the lives of people world-wide.

The lives of Japanese children were no exception. On 27 February 2020, the Prime Minister of Japan abruptly requested that elementary, junior high, high, special, and other schools nationwide be closed temporarily. From that day onwards, children were unable to attend school and meet with their friends and teachers. In Japan, the new school year begins in the month of April. Children, therefore, started a new school year without being able to organize their feelings. Even after June 2020, when schools were reopened in various areas, school life was forced to change completely. It is still in the midst of great turmoil, and the situation can be described as an emergency for the human population.

Such a situation will have a negative effect on children. UNICEF [2] warned that “this is a universal crisis and, for some children, the impact will be lifelong.” In addition, many emergency reports examined the effects of the COVID-19 pandemic on children’s bodies and minds. These studies have reported a decrease in physical activity [3,4,5,6], an increase in screen time [4,6,7], weight gain [8,9,10], decreased visual acuity (VA) [4,11,12] and mental health [4,13,14,15], and associated concerns. Studies on Japanese children have reported similar results; however, these were based on self-reports rather than actual measurements [16,17].

Article 13 of the School Health and Safety Law of Japan states that “at school, health check-ups must be conducted on a regular basis in every grade” [18]. Hence, the enforcement regulations for School Health and Safety Law require that health check-ups including height, weight, eyesight, hearing, dental caries, etc., be conducted between April and June every year [19]. However, in 2020, when the COVID-19 pandemic struck, the Ministry of Education, Culture, Sports, Science, and Technology communicated to each board of education and other relevant bodies that “if health check-ups cannot be carried out by the due date, for reasons such as the implementation system not being established due to the influence of the new coronavirus infection, it should be carried out as soon as possible by the end of the school year” [20]. Therefore, many schools conducted health check-ups several months later than usual. Comparing the actual results of the health check-ups in 2019 with those of 2020 demonstrates the effects of the COVID-19 pandemic, including temporary school closure, on children’s health. This may aid in providing evidence-based outcomes and discussions with respect to situations in Japan.

Myopia in East Asian children is already internationally considered a problem [21,22]. The number of Japanese children with poor visual acuity is also increasing [23]. The circumstances arising from the COVID-19 pandemic lifestyle—including temporary school closure and the request of national and local governments to limit outdoor activities—resulted in an increase in screen time (due to the increased use of video games, social networking services, and online learning platforms) and a decrease in outdoor activities among children. Thus, there is a concomitant concern regarding the effect of these pandemic-related lifestyle changes on myopia in Japanese children.

It is essential to urgently examine the consequences of the COVID-19 pandemic on children’s VA in order to determine its possible effects on children as they grow older. Our study aimed to examine the results of children’s VA tests from school health check-ups during the COVID-19 pandemic.

## 2. Materials and Methods

### 2.1. Ethics Approval

The study design was approved by the Ethics Committee of Nippon Sport Science University (approval no. 020-H002). The survey was conducted with the consent of the principals and staff of each school that participated in our study. Moreover, only anonymized data were collected.

### 2.2. Data Collection

We used snowball sampling to collect the VA test values from school health check-ups in 2019 and 2020 at each participating school. The study sample included 5893 children (3099 boys and 2794 girls) enrolled in seven public elementary schools (one city, four suburbs, and two agricultural and mountain villages) and four public junior high schools (two cities, one suburb, and one agricultural and mountain village) in Tokyo, Saitama, Kanagawa, and Shizuoka (Table 1).

### 2.3. VA Test and Assessment Method

We conducted all VA tests at each participating school in April 2019 and from May to July 2020. VA was measured and determined according to the standard methods used in school health check-ups in Japan. Briefly, a Landolt ring VA chart conforming to international standards was used. The test was conducted as follows. First, the distance from the optotype to the eyes was 5 m, and the participant was standing. Second, the height of the eyeball and the height of the target were almost equal, and the target was presented vertically such that the line of sight and the target surface were at right angles. Third, the left eye was shielded while ensuring that the eye shield did not press into the eye. We then requested the participant state where the break in the Landolt ring of the optotype was without squinting, using the right eye. The same procedure was carried out for the left eye. Fourth, as a general rule, the inspection was started from the 0.3 indicator. One of the four directions (up, down, left, and right) was presented for approximately 5 s. If the participant identified the direction correctly, the examination proceeded to the 0.7 and 1.0 indicators.

The test result was “answered correctly” when three or more of the four directions were correct, and “indeterminate” when two or fewer directions were correct. Therefore, a score ≥1.0 was classified as “A,” and a score between 0.7 and 1.0 as “B,” 0.3–0.7 as “C,” and <0.3 as “D.” Uncorrected VA tests were not conducted for children who wore eyeglasses or contact lenses, presuming their VA was poor. Therefore, they were classified as “E.” We defined B, C, D, and E as the poor VA conditions in our study.

### 2.4. Data Analysis

The following three points were analyzed based on the collected data.

First, we compared the assessment results of the VA tests of 2019 with those of 2020 according to the cross-sectional data. In this examination, the assessment results of both left and right eyes were compared, and the worst result was used as the VA value of each participant. The actual VA value of the participant was surveyed. Subsequently, the classifications B, C, D, and E were added, and the VA values in 2019 and 2020 were compared for each school stage using binomial logistic regression analysis. In the analysis, VA (0 = A, 1 = B to E) was entered as the dependent variable, survey year (0 = 2019, 1 = 2020) as the independent variable, and sex and grade as adjustment variables.

Second, we analyzed the changes in VA from 2019 to 2020, considering the individual differences by the longitudinal data. In this examination, binomial logistic regression analysis was performed with the change in VA from 2019 to 2020 (0 = no change or improvement, 1 = worse) as the dependent variable, VA in 2019 as the independent variable, and sex and grade as adjustment variables.

Third, we compared the 2019 VA laterality with that of 2020 by the longitudinal data. In this examination—excluding participants with vision correctors (E)—those who had the same assessment results for both eyes (e.g., A and A), those with a one-step difference (A and B, B and C, C and D), those with a two-step difference (A and C, B and D), and those with a three-step difference (A and D) were evaluated. Binomial logistic regression analysis was subsequently performed with the existence of laterality of one or more steps as the dependent variable (0 = no, 1 = yes), survey year (0 = 2019, 1 = 2020) as the independent variable, and sex and grade as adjustment variables.

For these analyses, we used all data for the first examination, except for those with vision correctors (E) and first graders in elementary and junior high school who were not tested in 2019. We used VA values for the second examination, and the data except for the corrected VA (E) were used for the third examination. IBM SPSS Statistics for Windows, Version 26.0 (IBM Corp., Armonk, NY, USA) was used for this series of statistical processing, and the significance level was set at *p* < 0.05.

## 3. Results

Table 2 shows the assessment results of the VA test according to sex and grade. In the first grade of the elementary school in 2019, 88.9% of boys and 84.1% of girls had A scores for their VA, i.e., they had good VA. In 2020, the percentage of A scores was 77.5% for boys and 75.3% for girls. However, it decreased as the grade increased, reaching 44.3% for boys and 32.0% for girls in 2019, and 39.1% for boys and 40.3% for girls in 2020.

Moreover, based on the actual VA condition, we examined the relationship between the survey year and poor VA using binomial logistic regression analysis (Figure 1). A significant regression coefficient was observed for the survey year, with odds ratios (ORs) of 1.249 (95% confidence interval [CI], 1.087–1.435) for elementary school students, and 1.231 (95% CI, 1.037–1.462) for junior high school students.

We examined the relationship between the VA value in 2019 and the change in VA from 2019 to 2020 (Figure 2). We observed significant regression coefficients in elementary school students with “B” (OR, 3.422; 95% CI, 2.410–4.858) and “C” (OR, 2.634; 95% CI, 1.870–3.709) scores, and junior high school students with “B” (OR, 4.406; 95% CI, 2.756–7.043), “C” (OR, 3.946; 95% CI, 2.643–5.893), and “D” (OR, 2.098; 95% CI, 1.376–3.199) scores.

Table 3 shows the laterality of VA by the sex and grade of the participants. In 2019, 5.6–20.9% of boys and 9.9–25.2% of girls had left-right VA differences. In 2020, these rates were 14.5–26.4% for boys and 16.0–30.6% for girls.

The results of the analysis of the relationship between the survey year and the laterality of VA are shown in Figure 3. We observed a significant regression coefficient in the survey year, with ORs of 1.516 (95% CI, 1.265–1.818) for elementary school students, and 1.423 (95% CI, 1.136–1.782) for junior high school students.

## 4. Discussion

We conducted the study with children from a range of different areas—urban, residential, and rural—and their VA measurements were obtained from school health check-ups and then analyzed. It was confirmed that there were more children with poor VA in 2020, the beginning of the COVID-19 pandemic, than in 2019.

According to the School Health Statistics Research conducted by the Ministry of Education, Culture, Sports, Science, and Technology, the results of VA assessments in the classes with even one corrected VA were not aggregated; therefore, those results cannot be compared with the results of our study, for which the class information of the participants was not available. Based on the data from the School Health Statistics of Tokyo in 2020 [24], which described the ratio of examiners only for corrected VA, we compared the percentage of children with an unaided VA of <1.0. It was confirmed that the number of participants with an unaided VA of <1.0, a so-called “poor VA,” was rather smaller in our study.

Living in urban areas has been identified as an environmental factor that promotes the progression of myopia [25]. For this reason, it can be presumed that the difference in the target area between the previous study (targeting only children in an urban area, Tokyo) and our study (targeting children in several different types of areas, including Tokyo) was reflected in the differences in the poor VA of our study. Therefore, it can be considered that the participants of our study were Japanese children who demonstrated VA values that could be generalized nation-wide; however, considering that children in East Asia, including Japan, may demonstrate poor VA [21,22], we deemed that it was a reality that cannot be overlooked.

Recently, numerous reports have suggested that outdoor activity is involved in suppressing myopia [26,27,28,29,30], and its effectiveness for doing so has been demonstrated in a prospective randomized trial comparing intervention and control groups that encouraged outdoor activity during breaks [31]. It has been reported that the COVID-19 pandemic not only increased the screen time that children engaged in [6,7] but also decreased the amount of physical activity [3,4], which is a reflection of outdoor activities. This also applies to the results of an emergency survey conducted on Japanese children during the temporary closure of schools [16,17]. In addition, a result from a sub-national survey among children and adolescents (aged 6–17 years) belonging to schools in Tokyo also supported the finding that the changes in the rate of poor VA were greater between 2019 and 2020 (during the pandemic) compared to the trends for the pre-pandemic period (between 2000 and 2019) [24]. The above survey and the findings of our study indicate that the number of children with poor VA increased in 2020; thus, it can be inferred that the decrease in VA in 2020 was caused by the pandemic. This finding raises important considerations for the rapidly advancing digitization of education.

Further, we demonstrated that children who already had poor VA had a greater tendency towards a more significant deterioration in 2020. Similarly, various studies have reported that myopic progression is related to the degree of myopia [32,33], whereas others have pointed out that the number of cases of severe myopia increases with the increase in early-onset cases [34]. By extrapolation, the effects of the lifestyle changes brought about by the COVID-19 pandemic had an even more significant effect on children who already had poor VA.

Depth perception is based on the deviation between the retinal images of the left and right eyes, and through this mechanism, a 3D sensation is obtained. Hu et al. [35] described the prevalence and associations of anisometropia in 6025 (94.7%) of 6364 eligible children aged 4–18 years. They reported that refractive anisometropia and anisomyopia increased with systemic parameters, including age, parental education level, and children’s lifestyle. The incidence of anisometropia and anisomyopia increased with an increase in the time spent indoors reading or writing. Moreover, Golebiowski et al. [36] indicated that the binocular accommodative facility decreased from a median of 11.3 to 7.8 cycles/min after smartphone use. Other studies have reported that binocular vision is associated with specific learning disabilities [37,38] and impaired ball game skills [39,40]. Although we did not examine the refractive error among children, the left-right differences in VA can be considered a reflection of anisometropia and binocular vision. Therefore, it can be speculated that the increase in near work, such as screen time, supports the findings of our study that the left-right difference in VA worsened in 2020.

In an emergency such as the COVID-19 pandemic, people tend to pay attention to delays in children’s learning, and they give importance to adequate learning even if it is under stressful conditions. However, schools exist for not only learning but also the health and growth of children. Many studies have suggested that childhood health is important for lifelong health. Moreover, without a healthy body and mind, effective and sustainable learning cannot be guaranteed. The United Nations Committee on the Rights of the Child [41] has also warned that the social, educational, economic, and recreational impacts of a pandemic on children’s rights should be considered. Therefore, the findings from our study support evidence-based education by schools during the pandemic.

Finally, although we have reported valuable findings which require further investigation, our study has several limitations. First, we should consider the difference in inspection months between 2019 and 2020. Vision loss progresses rapidly in children of the age group targeted in our study, as evidenced by the results presented in Table 2. Therefore, even with a delay of only one or two months, the delay in conducting health check-ups might have affected the findings. Second, we did not collect myopia and anisometropia data based on the degree of refraction and visual functions other than static VA, although it can be inferred that these test values were also affected by the pandemic. Therefore, it cannot be determined whether the poor VA was due to a structural abnormality. For example, in the case of children, although not many, there are cases where the results of the VA test deteriorate due to psychological effects. Third, only the VA values of school health check-ups were analyzed and not any other measures; however, they might also have been affected by the pandemic-related lifestyle changes. Since the result of the Japanese School Health Statistics Research used in our study did not have any similar counterparts in other countries, it may be considered an international property. For this reason, we also believe that Japan has the social responsibility to examine the effects of the COVID-19 pandemic more broadly, not limited to visual acuity, based on the results of the school health check-ups in Japan. These limitations may form the basis for future research topics.

## 5. Conclusions

Compared with 2019 data obtained before the start of the pandemic, data obtained during the pandemic in 2020 shows that more children had poor VA and laterality of VA, with a worsening of VA in children who already had poor vision. We believe that the lifestyle during the COVID-19 pandemic could have affected the VA and eye laterality of children. Hence, we suggest prioritizing the eye health of children and placing a greater emphasis on monitoring and preventing the progression of myopia. Interventions should include recommendations for limiting screen time (video games, etc.), and/or improving green time (outdoor play, etc.). In addition to the above, we conclude that these recommendations are also important for the future of society where the arrival of a new era of artificial intelligence is expected.

## Figures and Tables

**Figure 1 children-09-00342-f001:**
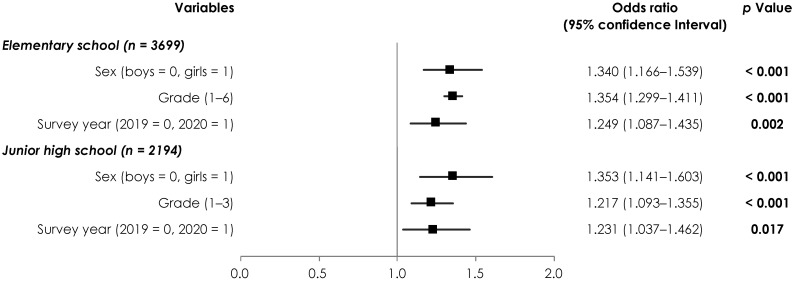
Binomial logistic regression analysis of the relationship between the survey year and poor eyesight. The dependent variable was eyesight score (0 = A, 1 = B–E), and the independent variable was the survey year. Measures of associations are displayed as odds ratios (black squares) and 95% confidence intervals (horizontal spikes). Significant *p* values are shown in bold.

**Figure 2 children-09-00342-f002:**
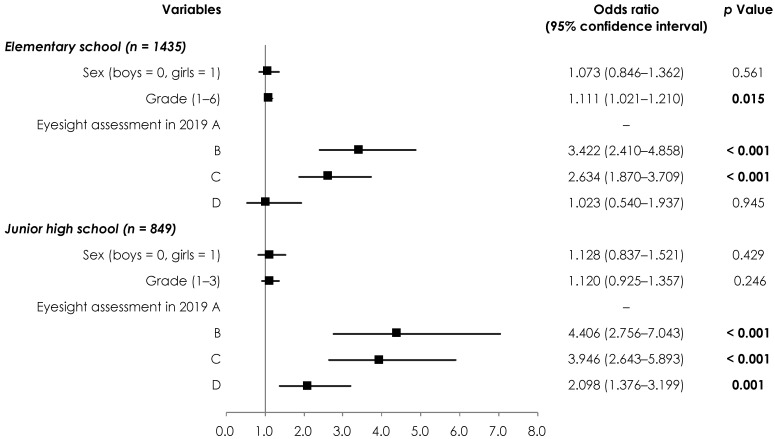
Relationship between eyesight in 2019 and the changes in eyesight from 2019 to 2020. The dependent variable was the change in eyesight from 2019 to 2020 (0 = no change or positive change, 1 = negative change), and the independent variable was the eyesight in 2019. Measures of associations are displayed as odds ratios (black squares) and 95% confidence intervals (horizontal spikes). Significant *p* values are shown in bold.

**Figure 3 children-09-00342-f003:**
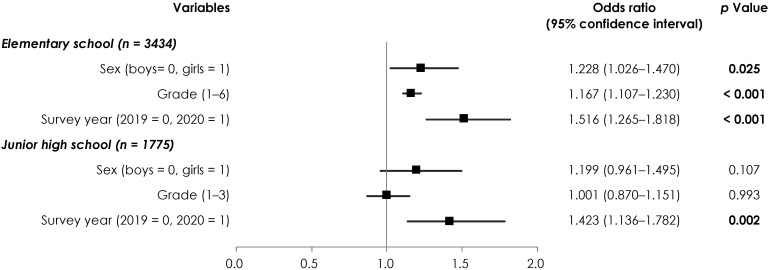
Relationship between the survey year and the left-right difference in eyesight. The dependent variable was the left-right difference in eyesight (0 = same, 1 = different), and the independent variable was the survey year. Measures of associations are displayed as odds ratios (black squares) and 95% confidence intervals (horizontal spikes). Significant *p*-values are shown in bold.

**Table 1 children-09-00342-t001:** Descriptive characteristics of participants and participating schools.

Participants (Total: *n* = 5893)	
Sex Grade	Boys: 3099, Girls: 2794 ES 1: 664, ES 2: 610, ES 3: 576, ES 4: 613, ES 5: 625, ES 6: 611, JHS 1: 859, JHS 2: 776, JHS 3: 559
**Survey school** **(Total: *n* = 11)**	
School classification School district School closure period Measurement month in 2020	ES: 7, JHS: 4 City: 3, Suburb: 5, Rural: 3 <80 days: 2, <90 days: 3, <100 days: 6 May: 1, June: 8, July: 2

Note: ES = elementary school; JHS = junior high school; *n* = number.

**Table 2 children-09-00342-t002:** Assessment results of the VA test according to sex and grade.

		A (1.0 ≤ A)		B (0.7 ≤ B < 1.0)		C (0.3 ≤ C < 0.7)		D (D < 0.3)		E (Vision Correctors)	
		2019	2020	2019	2020	2019	2020	2019	2020	2019	2020
Boys	ES 1 ES 2 ES 3 ES 4 ES 5 ES 6 JHS 1 JHS 2 JHS 3	144 (88.9) 120 (75.9) 97 (67.8) 116 (69.0) 109 (66.9) 83 (50.3) 125 (59.8) 107 (49.8) 39 (44.3)	124 (77.5) 124 (76.5) 100 (63.3) 76 (53.1) 91 (54.2) 86 (52.8) 131 (52.6) 103 (49.0) 84 (39.1)	13 (8.0) 13 (8.2) 13 (9.1) 13 (7.7) 15 (9.2) 14 (8.5) 22 (10.5) 13 (6.0) 12 (13.6)	19 (11.9) 15 (9.3) 18 (11.4) 18 (12.6) 19 (11.3) 20 (12.3) 26 (10.4) 24 (11.4) 20 (9.3)	3 (1.9) 17 (10.8) 23 (16.1) 20 (11.9) 18 (11.0) 22 (13.3) 20 (9.6) 39 (18.1) 18 (20.5)	12 (7.5) 15 (9.3) 23 (14.6) 32 (22.4) 27 (16.1) 25 (15.3) 29 (11.6) 28 (13.3) 29 (13.5)	0 (0.0) 2 (1.3) 3 (2.1) 2 (1.2) 6 (3.7) 19 (11.5) 15 (7.2) 28 (13.0) 14 (15.9)	3 (1.9) 5 (3.1) 9 (5.7) 3 (2.1) 10 (6.0) 18 (11.0) 13 (5.2) 22 (10.5) 25 (11.6)	2 (1.2) 6 (3.8) 7 (4.9) 17 (10.1) 15 (9.2) 28 (16.4) 27 (12.9) 28 (13.0) 5 (5.7)	2 (1.3) 3 (1.9) 8 (5.1) 14 (9.8) 21 (12.5) 14 (8.6) 50 (20.1) 33 (15.7) 57 (25.5)
Girls	ES 1 ES 2 ES 3 ES 4 ES 5 ES 6 JHS 1 JHS 2 JHS 3	138 (84.1) 87 (69.0) 94 (63.1) 85 (55.2) 74 (53.2) 62 (43.1) 80 (47.1) 91 (50.3) 24 (32.0)	134 (75.3) 119 (72.6) 79 (62.7) 80 (54.1) 73 (47.1) 57 (41.0) 93 (40.3) 65 (38.2) 73 (40.3)	16 (9.8) 21 (16.7) 21 (14.1) 16 (10.4) 11 (7.9) 20 (13.9) 12 (7.1) 17 (9.4) 4 (5.3)	22 (12.4) 22 (13.4) 18 (14.3) 12 (8.1) 19 (12.3) 16 (11.5) 27 (11.7) 14 (8.2) 21 (11.6)	6 (3.7) 12 (9.5) 19 (12.8) 28 (18.2) 19 (13.7) 24 (16.7) 18 (10.6) 23 (12.7) 14 (18.7)	12 (6.7) 15 (9.1) 16 (12.7) 27 (18.2) 31 (20.0) 21 (15.1) 26 (11.3) 22 (12.9) 23 (12.7)	2 (1.2) 3 (2.4) 6 (4.0) 11 (7.1) 19 (13.7) 21 (14.6) 18 (10.6) 27 (14.9) 23 (30.7)	2 (1.1) 5 (3.0) 6 (4.8) 13 (8.8) 16 (10.3) 27 (19.4) 34 (14.7) 22 (12.9) 18 (9.9)	2 (1.2) 3 (2.4) 9 (6.0) 14 (9.1) 16 (11.5) 17 (11.8) 42 (24.7) 23 (12.7) 10 (13.3)	8 (4.5) 3 (1.8) 7 (5.6) 16 (10.8) 16 (10.3) 18 (12.9) 51 (22.1) 47 (27.6) 46 (25.4)

Note: The results in the table indicate number (%); VA=visual acuity; ES = elementary school; JHS = junior high school.

**Table 3 children-09-00342-t003:** Left-right difference in eyesight by sex and grade.

		Same		One-Step Difference		Two-Step Difference		Three-Step Difference	
		2019	2020	2019	2020	2019	2020	2019	2020
Boys	ES 1 ES 2 ES 3 ES 4 ES 5 ES 6 JHS 1 JHS 2 JHS 3	151 (94.4) 130 (85.5) 120 (88.2) 134 (88.7) 131 (88.5) 115 (83.3) 149 (81.9) 148 (79.1) 66 (79.5)	135 (85.4) 136 (85.5) 122 (81.3) 95 (73.6) 114 (77.6) 114 (76.5) 155 (77.9) 132 (74.6) 119 (75.8)	8 (5.0) 19 (12.5) 12 (8.8) 14 (9.3) 14 (9.5) 17 (12.3) 23 (12.6) 33 (17.6) 15 (18.1)	22 (13.9) 20 (12.6) 23 (15.3) 28 (21.7) 27 (18.4) 27 (18.1) 33 (16.6) 42 (23.7) 30 (19.0)	1 (0.6) 3 (2.0) 4 (2.9) 3 (2.0) 2 (1.4) 4 (2.9) 6 (3.3) 5 (2.7) 2 (2.4)	1 (0.6) 3 (1.9) 4 (2.7) 6 (4.7) 6 (4.1) 7 (4.7) 10 (5.0) 0 (0.0) 6 (3.8)	0 (0.0) 0 (0.0) 0 (0.0) 0 (0.0) 1 (0.7) 2 (1.4) 4 (2.2) 1 (0.5) 0 (0.0)	0 (0.0) 0 (0.0) 1 (0.7) 0 (0.0) 0 (0.0) 1 (0.7) 1 (0.5) 2 (0.8) 2 (1.3)
Girls	ES 1 ES 2 ES 3 ES 4 ES 5 ES 6 JHS 1 JHS 2 JHS 3	146 (90.1) 105 (85.4) 123 (87.9) 110 (78.6) 99 (80.5) 95 (74.8) 97 (75.8) 129 (81.6) 53 (81.5)	142 (83.5) 134 (83.2) 100 (84.0) 105 (79.5) 103 (74.1) 89 (73.6) 125 (69.4) 90 (73.2) 95 (70.4)	15 (9.3) 16 (13.0) 15 (10.7) 24 (17.1) 15 (12.2) 27 (21.3) 23 (18.0) 22 (13.9) 10 (15.4)	22 (12.9) 22 (13.7) 18 (15.1) 25 (18.9) 31 (22.3) 24 (19.8) 41 (22.8) 23 (18.7) 25 (18.5)	1 (0.6) 2 (1.6) 1 (0.7) 6 (4.3) 8 (6.5) 5 (3.9) 6 (4.7) 3 (1.9) 1 (1.5)	6 (3.5) 5 (3.1) 1 (0.8) 1 (0.8) 3 (2.2) 4 (3.3) 11 (6.1) 7 (5.7) 10 (7.4)	0 (0.0) 0 (0.0) 1 (0.7) 0 (0.0) 1 (0.8) 0 (0.0) 2 (1.6) 4 (2.5) 1 (1.5)	0 (0.0) 0 (0.0) 0 (0.0) 1 (0.8) 2 (1.4) 4 (3.3) 3 (1.7) 3 (2.4) 5 (3.7)

Note: ES = elementary school; JHS = junior high school; *n* = number.

## Data Availability

The data that supports the findings of this study are available from the corresponding author, upon reasonable request.

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
