# Peer review of "The Changes in Visual Acuity Values of Japanese School Children during the COVID-19 Pandemic"

_children, 2022, doi:10.3390/children9030342_

Round 1
Reviewer 1 Report
Congratulation for the article.
The article sent for review analyzes the quality visual acuity before and during the COVID-19 pandemic.
The article demonstrates how important is the visual screening of students.
The article provides a good statistical analysis and uses the academic language.
Because testing the participants consisted only in measuring visual acuity, other correlations cannot be made.
Author Response
Thank you very much for your time. We are thankful for the time and energy you expended.
Reviewer 2 Report
This is a well-written paper looking at school visual acuity testing results in Japan, comparing 2019 to late 2020 to determine the effect of the COVID-19 pandemic.
1. I assume that uncorrected visual acuity is what was tested (given group E were those wearing glasses and contacts). Please explicitly state this in the methods.
2. "Poor visual acuity" should be defined in the methods. It actually may be better to refer to this as "reduced visual acuity" because anything less than 1.0 appears to be considered "poor", though clinically we would not consider this "poor" but rather "reduced".
3. It would be a better paper if you could include data from at least 1 prior year (2018?) to show that this significant change is really due to the COVID-19 pandemic rather than a general worsening year-by-year.
4. Lines 221-222: "Further, we demonstrated that children who already had poor VA had a greater tendency towards a more significant deterioration in 2020." I am not sure this follows from the study. I was under the impression that cross-sectional data were used, rather than individual longitudinal data. Please explain methods better or change this sentence to read more like a comparison of cross-sectional data from each year. If longitudinal data are used, then certainly additional years of study are needed to see if there is a difference between 2019-2020 comparison vs. 2018-2019 or 2017-2018, etc.
5. Please specify in Methods use of cross-sectional data vs. longitudinal data.
6. The discussion of limitations of the study is well-written.
Author Response
Thank you very much for your time. We are thankful for the time and energy you expended. Receiving your comments, we revised as belows.
Reviewer 2 |
|
This is a well-written paper looking at school visual acuity testing results in Japan, comparing 2019 to late 2020 to determine the effect of the COVID-19 pandemic. |
|
1. I assume that uncorrected visual acuity is what was tested (given group E were those wearing glasses and contacts). Please explicitly state this in the methods. |
As you pointed out, we thought it was unclear. Therefore, we added "uncorrected" in L109-110. |
2. "Poor visual acuity" should be defined in the methods. It actually may be better to refer to this as "reduced visual acuity" because anything less than 1.0 appears to be considered "poor", though clinically we would not consider this "poor" but rather "reduced". |
Thanks for your important comments. We thought it need the definition of poor VA. Therefore, we added it in L110-111. However, for example, poor VA has a mixture of those who have reduced to "D" and those who were originally "D." Hence, we thought that "poor" which means state, is more appropriate than "reduce" which means change. |
3. It would be a better paper if you could include data from at least 1 prior year (2018?) to show that this significant change is really due to the COVID-19 pandemic rather than a general worsening year-by-year.
|
Thank you for the valuable advice. Unfortunately, we don't have 2018 data in our sample. But we confirmed that the changes in the rate of VA were greater between 2019 and 2020 (during pandemic) compared to the trend for pre-pandemic, which was conducted by the Tokyo Metropolitan Board of Education. Therefore, we added this information in discussion (L218-222.) Please find new reference 24. |
4. Lines 221-222: "Further, we demonstrated that children who already had poor VA had a greater tendency towards a more significant deterioration in 2020." I am not sure this follows from the study. I was under the impression that cross-sectional data were used, rather than individual longitudinal data. Please explain methods better or change this sentence to read more like a comparison of cross-sectional data from each year. If longitudinal data are used, then certainly additional years of study are needed to see if there is a difference between 2019-2020 comparison vs. 2018-2019 or 2017-2018, etc. |
Thanks for pointing this out. We thought it was imperfect describe. So, we added to the "data analysis" (L117, 125, 129.) |
5. Please specify in Methods use of cross-sectional data vs. longitudinal data. |
|
6. The discussion of limitations of the study is well-written. |
|
For all your figure, i am not sure dot with confidence intervalle is usefull. Instead, you can build a table with the percentage and OR |
We did include actual numbers for OR and 95%CI within the figures. We believe that the information allows readers to understand the actual values of OR and 95% CI. |
Reviewer 3 Report
dear author
thank you for your manuscript, please change letters A, B, C by the VA they mean because it is confusing through the article and especially in the abstract
as i understand, in case of decrease in VA, you cannot state if it is due to a limit or to refraction error. Maybe, you can state it clearlly
For all your figure, i am not sure dot with confidence intervalle is usefull. Instead, you can build a table with the percentage and OR
best regards
Author Response
Thank you very much for your time. We are thankful for the time and energy you expended. Receiving your comment, we revised as belows.
Reviewer 3 |
|
thank you for your manuscript, please change letters A, B, C by the VA they mean because it is confusing through the article and especially in the abstract |
Thanks for your important comments. We thought it was unclear. Therefore, we added it in L25-26.
|
as i understand, in case of decrease in VA, you cannot state if it is due to a limit or to refraction error. Maybe, you can state it clearlly
|
We didn’t collect the data of degree of refraction. Therefore, it cannot be determined whether the poor VA is due to a structural abnormality. For example, in the case of children, although not many, there are cases where the results of the VA test deteriorate due to psychological effects. From the above, we decided to write as limitation. |
For all your figure, i am not sure dot with confidence intervalle is usefull. Instead, you can build a table with the percentage and OR
|
We did include actual numbers for OR and 95%CI within the figures. We believe that the information allows readers to understand the actual values of OR and 95% CI.
|
Round 2
Reviewer 3 Report
dear authors
thank you for modifications
for limitation about refraction, please state clearly what you wrote in your cover letter
We didn’t collect the data of degree of refraction. Therefore, it cannot be determined whether the poor VA is due to a structural abnormality. For example, in the case of children, although not many, there are cases where the results of the VA test deteriorate due to psychological effects.
best regards
Author Response
Thanks for your important comment. We thought that you were right. Therefore, we added it in L263-265. Please confirm it.
